# Susceptibility of *Aerococcus urinae* and *Aerococcus sanguinicola* to Standard Antibiotics and to Nitroxoline

Aysel Ahmadzada,[a] Frieder Fuchs,[a,b] Axel Hamprecht[a,c,d]

[a]Institute for Medical Microbiology, Immunology and Hygiene, University of Cologne, Medical Faculty and University Hospital of Cologne, Cologne, Germany
[b]Department of Microbiology and Hospital Hygiene, Bundeswehr Central Hospital Koblenz, Koblenz, Germany
[c]German Centre for Infection Research (DZIF), Bonn-Cologne, Cologne, Germany
[d]Institute for Medical Microbiology and Virology, University of Oldenburg, Oldenburg, Germany

Aysel Ahmadzada and Frieder Fuchs contributed equally to this work. Author order was determined alphabetically.

**ABSTRACT** *Aerococcus urinae* and *Aerococcus sanguinicola* have been increasingly recognized as causative agents of urinary tract infection (UTI) during the last decade. Nitroxoline achieves high urinary concentrations after oral administration and is recommended in uncomplicated UTI in Germany, but its activity against *Aerococcus* spp. is unknown. The aim of this study was to assess the *in vitro* susceptibility of clinical *Aerococcus* species isolates to standard antibiotics and to nitroxoline. Between December 2016 and June 2018, 166 *A. urinae* and 18 *A. sanguinicola* isolates were recovered from urine specimens sent to the microbiology laboratory of the University Hospital of Cologne, Germany. Susceptibility to standard antimicrobials was analyzed by disk diffusion (DD) according to EUCAST methodology, nitroxoline was tested by DD and agar dilution. Susceptibility of *Aerococcus* spp. to benzylpenicillin, ampicillin, meropenem, rifampicin, nitrofurantoin, and vancomycin was 100% and resistance was documented only against ciprofloxacin (20 of 184; 10.9%). MICs of nitroxoline in *A. urinae* isolates were low (MIC$_{50/90}$ 1/2 mg/L) while significantly higher MICs were observed in *A. sanguinicola* (MIC$_{50/90}$ 64/128 mg/L). If the EUCAST nitroxoline breakpoint for *E. coli* and uncomplicated UTI was applied (16 mg/L), 97.6% of *A. urinae* isolates would be interpreted as susceptible while all *A. sanguinicola* isolates would be considered resistant. Nitroxoline demonstrated high activity against clinical *A. urinae* isolates, but low activity against *A. sanguinicola*. Nitroxoline is an approved antimicrobial for UTI and could be an alternative oral drug to treat *A. urinae* urinary tract infection, yet clinical studies are needed to demonstrate this potential *in vivo*.

**IMPORTANCE** *A. urinae* and *A. sanguinicola* have been increasingly recognized as causative agents in urinary tract infections. Currently, there are few data available on the activity of different antibiotics against these species and no data on nitroxoline. We demonstrate that clinical isolates in Germany are highly susceptible to ampicillin, while resistance to ciprofloxacin was common (10.9%). Additionally, we show that nitroxoline is highly active against *A. urinae*, but not against *A. sanguinicola*, which based on the presented data, should be considered intrinsically resistant. The presented data will help to improve the therapy of urinary tract infections by *Aerococcus* species.

**KEYWORDS** urinary tract infection, uUTI, intrinsic resistance, oral, MDR, 8-hydroxy-quinoline

Address correspondence to Axel Hamprecht, axel.hamprecht@uol.de.

The authors declare no conflict of interest.

*A*erococcus spp. are fastidious Gram-positive cocci and have been recognized as emerging microbes in urinary tract infections (UTI), bacteremia, and endocarditis (1, 2). Matrix-assisted laser desorption ionization time-of-flight mass spectrometry (MALDI-TOF MS) has facilitated the identification and discrimination of *A. urinae* and

**TABLE 1** Susceptibility of *Aerococcus* spp.

| Antibiotic | Susceptibility | |
| --- | --- | --- |
| | *A. urinae* | *A. sanguinicola* |
| Ampicillin[a] | 100% (166/166) | 100% (18/18) |
| Meropenem[a] | 100% (166/166) | 100% (18/18) |
| Ciprofloxacin[a] | 90.4% (150/166) | 77.8% (14/18) |
| Rifampicin[a] | 100% (166/166) | 100% (18/18) |
| Vancomycin[a] | 100% (166/166) | 100% (18/18) |
| Nitrofurantoin[a] | 100% (166/166) | 100% (18/18) |
| Nitroxoline[b] | 97.6% (162/166) | 0% (0/18) |

[a]Interpreted according to EUCAST disk diffusion breakpoints for *A. urinae* and *A. sanguinicola*.
[b]Interpreted according to EUCAST breakpoints for *E. coli* ($\leq 16$ mg/L).

*A. sanguinicola* from other alpha-hemolytic species with similar colony morphology (3–5). Hence, the understanding of the significance of *Aerococcus* spp. in UTI has greatly improved. Additionally, reports from invasive infections with identification of aerococci as causative pathogens are increasing (6–8).

In 2017, EUCAST introduced clinical breakpoints for *A. sanguinicola* and *A. urinae*, which now encompass benzylpenicillin, ampicillin, meropenem, ciprofloxacin, levofloxacin, vancomycin, rifampicin, and nitrofurantoin. However, for UTI treatment several limitations are obvious: Except for nitrofurantoin, none of the antimicrobials with defined breakpoints is currently recommended for oral empirical treatment of lower UTI. Benzylpenicillin and ampicillin are only useful for targeted therapy, similarly for fluoroquinolones, vancomycin, rifampicin, or carbapenems, which are usually avoided because of side effects and collateral damage. Susceptibility data for *Aerococcus* spp. is still scarce and resistance to currently available oral antimicrobials (such as fluoroquinolones and nitrofurantoin) has been reported, highlighting the need for alternative options (9).

Nitroxoline is a quinoline derivate (8-hydroxy-quinoline) that is structurally unrelated to any other antibiotic and was first described as an antimicrobial in 1954 (10). It has been rediscovered for the treatment of uncomplicated UTI (uUTI) and is recommended as a first-line drug in the German uUTI guideline (11). The mode of action is based on ion chelation with subsequent effects on enzymatic pathways and charges of cellular compartments of fungal and bacterial cells (12). Recently, activity of nitroxoline against emerging pathogens such as multidrug resistant Gram-negative pathogens, including carbapenemase producers, enterococci, staphylococci, and even drug resistant fungi and mycobacteria was demonstrated (13–19). High urine concentrations can be achieved after oral administration (standard dose 250 mg every 8 h) yet systemic concentrations are low, limiting its use to infections of the urinary tract (20).

The aim of the present study was to compare the susceptibility of *A. urinae* and *A. sanguinicola* clinical isolates from Germany according to EUCAST and to assess the potential of nitroxoline for the treatment of *Aerococcus* spp.

## RESULTS

All aerococcal isolates tested in this study were susceptible to ampicillin, meropenem, vancomycin, nitrofurantoin, and rifampicin (Table 1). Resistance to ciprofloxacin was 16 of 166 (9.6%) in *A. urinae* and 4 of 18 (22.2%) in *A. sanguinicola*, $P = 0.11$ (Table 1).

MICs of nitroxoline in *A. urinae* isolates were low (MIC$_{50/90}$ 1/2 mg/L, MIC range 0.25 to 64 mg/L, inhibition zone diameter (IZD) range 12 to 30 mm) while significantly higher MICs were observed in *A. sanguinicola* (MIC$_{50/90}$ 64/128 mg/L, MIC range 32 to 128 mg/L, IZD range 6 to 13 mm) (Fig. 1). If the EUCAST nitroxoline breakpoint for *E. coli* and uncomplicated UTI was applied (16 mg/L), 97.6% (162/166) of *A. urinae* isolates would be interpreted as susceptible, compared to 0% (0/18) of *A. sanguinicola* isolates, $P < 0.0001$. Similarly, if disk diffusion inhibition zones were categorized using the EUCAST nitroxoline breakpoint for *E. coli* (15 mm), 163/166 (98.2%) of *A. urinae* isolates would be interpreted

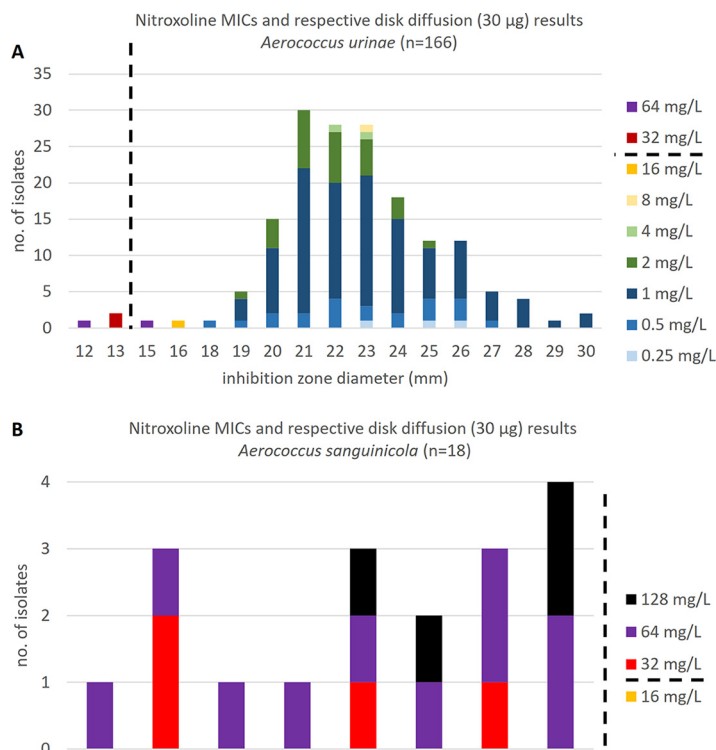

**FIG 1** Histograms showing MICs of clinical *A. urinae* (A) and *A. sanguinicola* (B) isolates versus respective inhibition zone by disk diffusion. Dashed line indicates EUCAST recommended breakpoints for *E. coli*.

as susceptible and 3/166 (1.8%) as resistant (Fig. 1). Additional information on correlation of disk diffusion and MICs is provided in Fig. S1 and S2.

## DISCUSSION

Urinary tract infections are a frequent cause for prescription of antibiotics in inpatients and outpatients, yet treatment has become more and more difficult with increasing antimicrobial resistance, especially in Gram-negative bacilli. Additionally, limited oral options are available for emerging microbes such as *Aerococcus* spp. (21). Moreover, the use of highly active agents, such as fluoroquinolones, has been restricted because of potential side effects (22, 23).

Therefore, old drugs such as nitroxoline, nitrofurantoin, and amdinocillin (mecillinam) have regained interest recently, since they show high activity against drug resistant pathogens and achieve high urinary concentrations after oral administration (11–20, 24, 25). Additionally, they are well tolerated, and ample experience is available with these drugs in different countries. Oskooi et al. (26) investigated aerococcal urinary tract infections and reported clinical success in all patients (100%) treated with amdinocillin, though microbiological success was only 33%. In contrast, the activity of nitroxoline against *A. urinae* has not been investigated previously.

Our study demonstrates the excellent *in vitro* activity of nitroxoline against *A. urinae* isolated from urinary specimens in a tertiary care center in Germany. If the current EUCAST breakpoint for *E. coli* (≤16 mg/L) was applied, 97.6% of *A. urinae* isolates would be considered susceptible to nitroxoline. Since oral antimicrobial options are limited for the treatment of *A. urinae*, nitroxoline could therefore be a useful alternative agent.

In contrast, activity of nitroxoline against *A. sanguinicola* was poor, indicating intrinsic resistance of this species to nitroxoline, which was demonstrated in both disk diffusion

and agar dilution. However, the small number of *A. sanguinicola* isolates limits the generalizability of this finding.

In another study, Scholtz et al. (27) demonstrated high susceptibility (97.8% to 100%) of *Aerococcus* spp. to benzylpenicillin, meropenem, linezolid, and vancomycin, but comparative lower susceptibility (56%) to levofloxacin on isolates from the United States. In this study, MICs were interpreted according to CLSI M45 breakpoints. In total, 134 *Aerococcus* isolates were tested using different commercially available susceptibility testing methods (gradient tests with different agars, Vitek2, BD Phoenix) and compared to broth microdilution as the CLSI-recommended reference method. The highest agreement with broth microdilution was observed for gradient tests on Mueller-Hinton agar with 5% sheep blood. In our study, an additional method was assessed (EUCAST disk diffusion), equally demonstrating high susceptibility of aerococci to standard antimicrobials when EUCAST breakpoints were applied, but a lower rate of resistance to fluoroquinolones (10.9%).

In the study of Roy et al. (28) on isolates from Canada, ciprofloxacin MICs $\geq$ 2 mg/L were recorded in 17.4% of isolates (I or R according to CLSI), which was more similar to the results from our study.

The present study investigated nitroxoline as an alternative treatment option with two different methodologies (disk diffusion and agar dilution). Overall correlation of inhibition zones and MICs was good, since isolates with high nitroxoline MICs showed small inhibition zones. However, only few *A. urinae* isolates with high nitroxoline MICs were available (4/166 isolates > 16 mg/L) and no *A. sanguinicola* isolate demonstrated low nitroxoline MICs, limiting interpretation of the correlation of the two methods. Also, more data on different MIC testing methods (e.g., agar dilution versus broth microdilution) for *Aerococcus* spp. are needed since superiority of agar dilution (as used in this study) was demonstrated for fluoroquinolones but not for other drugs previously (21, 28).

Nitroxoline resistance is still poorly understood and has rarely been described in clinical isolates. Molecular follow-up studies for comparison of genetic variations of *A. urinae* and *A. sanguinicola* should consider previously described mechanisms, such as efflux pumps (29).

Our study has several limitations. Not all isolates from the study period were available for analysis (e.g., were not stored initially or did not grow from frozen stock), which could limit the representativeness of our results. Furthermore, low MICs do not necessarily correlate with clinical or microbiological success *in vivo*. For nitroxoline, data on clinical outcome is still scarce and therapeutic failure has been described in geriatric patients for UTI caused by other organisms (30). Nevertheless, the activity displayed in the present study against *A. urinae* is promising, not only when applying the current EUCAST breakpoint (162/166 *A. urinae* $\leq$16 mg/L [97.6%]), but also when considering the pharmacokinetics of nitroxoline. High urinary concentrations after oral therapy have been demonstrated (5.4 mg/L of the unconjugated form and 210.6 mg/L of the conjugated form, yet the role of the conjugated form remains elusive) (20, 30). The fact that nitroxoline is an approved drug may facilitate *in vivo* studies and compassionate use, especially for uUTI.

Another limitation of the present study is the lack of clinical data, as we could not discriminate between asymptomatic bacteriuria and UTI as underlying conditions. This could lead to an overestimation of the uropathogenic potential of the isolates in our collection. Since most isolates were from inpatients from a tertiary care center, additional risk factors may be overrepresented compared to uncomplicated UTI for which nitroxoline is approved. Additionally, our patient population might not be comparable to other populations, e.g., from primary care, which could have an impact on the overall susceptibility. Although a recent study has investigated virulence factors of *Aerococcus* species isolated from UTI (31), more data on the correlation of clinical characteristics and outcome of aerococcal infections is needed.

To conclude, *A. urinae* isolates are highly susceptible against nitroxoline, in contrast to *A. sanguinicola* isolates, which can be considered intrinsically resistant. Nitroxoline

could be a useful alternative oral antibiotic for the treatment of uncomplicated UTI caused by *A. urinae.*

## MATERIALS AND METHODS

**Ethics approval.** We declare that the study was conducted in accordance with guidelines outlined by the Declaration of Helsinki. For this study on bacterial isolates obtained as part of routine laboratory diagnostics, no patient consent or ethical approval was required according to the regulations of the University Hospital of Cologne.

**Isolates.** All isolates were grown from urine samples, which were sent for bacteriological analysis to the Institute of Medical Microbiology of the University Hospital of Cologne, a large tertiary care center in the western part of Germany. Isolates were characterized as part of routine diagnostics, based on national microbiology quality standards (32) and stored at −80℃ in glycerol stocks. In the study period (December 2016 to June 2018), 30,718 urine samples from 17,570 patients were analyzed; of these, 312 grew *A. urinae/sanguinicola*. After removing duplicate isolates, 278 isolates remained. A total of 166 *A. urinae* and 18 *A. sanguinicola* nonduplicate isolates were available for our study, representing 66.2% of all *Aerococcus* spp. during the study period. Of these isolates, 82 were from voided midstream urine, 56 from catheterization, and 46 others. Most samples were from inpatients (121/184; 65.7% of isolates); of these, 93/184 (50,5%) were female and the median age was 77 years. For identification, MALDI-TOF MS (Microflex LT, Bruker Daltonik, Biotyper database 5.0, Bremen, Germany) was used. All isolates with a score ≥2.0 by Biotyper were included in the study.

**Antimicrobial susceptibility testing.** Susceptibility to ampicillin, meropenem, ciprofloxacin, vancomycin, and nitrofurantoin was determined by disk diffusion according to EUCAST methodology (21) on MH-F agar (Axonlab, Reichenbach, Germany). Antibiotic disks were obtained from i2a (Montpellier, France), except for nitroxoline 30 $\mu$g disks, which were purchased from MAST (Reinfeld, Germany).

Additionally, MICs were determined for nitroxoline using an agar dilution protocol for fastidious organisms, as previously described, and recommended by EUCAST (21, 28). Briefly, Mueller-Hinton (MH) agar was supplemented with defibrinated horse blood and other supplements according to EUCAST recommendations (33). Nitroxoline powder (Rosen Pharma, St.Ingbert, Germany) was dissolved in DMSO (Sigma-Aldrich, Darmstadt, Germany) and diluted into the agar plates in serial 2-fold dilutions to reach final concentrations of 0.06 mg/L to 128 mg/L. A suspension with an inoculum corresponding to $10^7$ CFU/mL was prepared, and 1 $\mu$L was inoculated on the agar plates using an automated multipoint inoculator. Reading was carried out after incubation at 35℃ for 24 h in a $CO_2$ enriched atmosphere. MICs were correlated with inhibition zone diameters (IZD) of disk diffusion testing.

**Statistical analysis.** Correlation of MICs and IZD was calculated by linear regression analysis using Excel (Microsoft, Redmond, WA, USA). Statistical analysis of categorical data were performed using Fisher's exact test (two-sided). A *P* value of <0.05 was considered significant.

**Data availability.** The original contributions presented in the study are included in the article/supplemental material. The corresponding author may be contacted for additional information.

## SUPPLEMENTAL MATERIAL

Supplemental material is available online only.

**SUPPLEMENTAL FILE 1**, PDF file, 0.6 MB.

## ACKNOWLEDGMENTS

We thank Marco Schwabe and Andrea Stammegna for their kind support in preparation of the agar plates with horse blood. Pure nitroxoline powder was provided free of charge by Rosen Pharma.

F.F. has a clinician scientist position supported by the Deans office, Medical Faculty, University of Cologne.

Investigation, Writing - Original Draft, Review & Editing, A.A.; Project Administration, Isolate Collection, Investigation, Writing – Original Draft, Writing – Review & Editing, F.F.; Conceptualization, Supervision, Validation, Resources, Writing – Review & Editing, A.H.

We declare no conflict of interest.

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
