## [Reviewer comments · Microbiology Spectrum]

Microbiology Spectrum

Susceptibility of *Aerococcus urinae* and *Aerococcus sanguinicola* to standard antibiotics and to nitroxoline

Aysel Ahmadzada, Frieder Fuchs, and Axel Hamprecht

Corresponding Author(s): Axel Hamprecht, University of Oldenburg

Review Timeline:

Submission Date:	July 20, 2022
Editorial Decision:	September 11, 2022
Revision Received:	November 24, 2022
Accepted:	February 5, 2023

Editor: Rebekah Martin

Reviewer(s): Disclosure of reviewer identity is with reference to reviewer comments included in decision letter(s). The following individuals involved in review of your submission have agreed to reveal their identity: Kristen Leigh Jurcic Smith (Reviewer #1); Erik Senneby (Reviewer #2); Magnus Rasmussen (Reviewer #4)

Transaction Report:

DOI: <https://doi.org/10.1128/spectrum.02763-22>

September 11, 2022

Prof. Axel Georg Hamprecht
University of Oldenburg
Institute for Medical Microbiology and Virology
Brandenburger Str. 19
Oldenburg 26133
Germany

Re: Spectrum02763-22 (Susceptibility of *Aerococcus urinae* and *Aerococcus sanguinicola* to standard antibiotics and to nitroxoline)

Dear Prof. Axel Georg Hamprecht:

Thank you for submitting your manuscript to Microbiology Spectrum. Your manuscript has been reviewed by experts in the field, and the consensus is that revisions are necessary. When submitting the revised version of your paper, please provide (1) point-by-point responses to the issues raised by the reviewers as file type "Response to Reviewers," not in your cover letter, and (2) a PDF file that indicates the changes from the original submission (by highlighting or underlining the changes) as file type "Marked Up Manuscript - For Review Only". Please use this link to submit your revised manuscript - we strongly recommend that you submit your paper within the next 60 days or reach out to me. Detailed instructions on submitting your revised paper are below.

Link Not Available

Sincerely,

Rebekah Martin

Journals Department
Reviewer comments:

Reviewer #1 (Comments for the Author):

1. Since a difference in susceptibilities was noted between *A. urinae* and *A. sanguinicola*, did you investigate the different patient populations? Were the individuals with *A. sanguinicola* treated with more antibiotics than the patients with *A. urinae*? This would help understand if the observed differences are intrinsic resistance vs. recent acquisitions (additional support for statement in Line 140)
2. Line 114 - for ciprofloxacin resistance, further discussion about observation in both *A. urinae* and *A. sanguinicola* could be beneficial.

3. As the authors note (Line 45), there is a need for additional susceptibility data in the literature. Could the authors add additional information on how the patients with *Aerococcus* sp. were treated? Is there correlation with the drugs that they studied and what was used for treatment?
4. Line 145 - to improve clarity, it would be helpful to compare and contrast methodology between the current submission and Scholtz et al.
5. Line 149 - "fluorochinolone" needs updated to "fluoroquinolone"
6. Discussion section should elaborate more on previous studies and how they are the same or different than present study (see comment 4).
7. See lines 150/151 - this paragraph is a single sentence and could be combined with 152 - 156.
8. Line 164 - comma not needed after "The fact"

Reviewer #2 (Comments for the Author):

The authors have performed AST on aerococcal isolates with a panel of antibiotics, including nitroxoline. To my knowledge, AST for nitroxoline and aerococci has not been reported before. There are some remarks and questions listed below.

L20 The authors write "...are emerging species..". Maybe it is more correct to say something like, "...have been increasingly recognized as cause of UTI during the recent decade".

L34 "all *A. urinae* isolates would be interpreted as susceptible". This seems to be incorrect according to the result (where 97.6 % were S).

L47 "while ciprofloxacin was more often resistant". This is incorrect. It is of course the isolates that are resistant, not the antibiotic.

L55-56 Same comment as on Line 20. The references used indicate that *Aerococcus* species have been recognized the last couple of years, but in fact have been increasingly recognized as cause of UTI during the recent decades. There are many references 5-15 years back that could be used also.

L59 I would suggest that there is a reference to the MALDI statement in the manuscript.

L64-66 A good and valid point.

L70 A better reference should be used to this statement.

L79 Administration instead of application?

L85 This section raises some questions. First, are the isolates in the study representative for all *A. urinae* and *A. sanguinicola* isolates? How many urine samples does the laboratory analyse each month/year? And what is the proportion of *Aerococcus* spp in the positive urine cultures? How were the samples selected - was it a random or a consecutive selection during the period? Which is the population base (size), is it both inpatient and outpatients? What were the gender and age distribution for all patients in the study? Was there any clinical information about symptoms?

L89 Which MALDI score was used as cut-off for reliable identification to the species level, and were all isolates identified according to this level?

L97 Reference 19 refers to Etest (EUCAST). Is that relevant for the method used?

L103 Temperature could also be specified.

L105 For what was the Mann-Whitney U test used?

L112 The sentence could be improved in line with "All aerococcal isolates tested in this study were susceptible to (Table 1)."

L119 The correlation coefficient (r-value) is not of great clinical interest. Also, there is a lack of discussion around what the result means and how it could be explained that the methods correlate rather poorly.

It would be of more value to see a histogram with MIC and zone diameter ("EUCAST model") and a proposed cut-off zone diameter for each species that could be used to categorize the isolates as S or R (when using MIC 16 mg/L).

Reviewer #3 (Comments for the Author):

The work that the authors have done in testing the susceptibility of *Aerococcus* to nitroxoline is interesting. There are however some potential issues that I believe should be addressed.

1. The collection of the studied samples is not clearly described. Were all findings of *Aerococcus urinae* and *A. sanguinicola* from 2016 to 2018 included in the study? If not, was there any possibility of a systematic bias in which samples were included or not? What percentage of samples identified during this period was included? Were all samples identified at the same

laboratory?

2. The patient population that the urinary samples were taken from is not described. Do the laboratory/ies in the study take samples from primary health care centers as well as secondary and tertiary care centers? From public or private clinics? This will potentially affect the population (with its accompanying microbiome) that is included in a study such as this. How large is the catchment area of the microbiology laboratory with its associated hospitals and other clinics? Knowledge of these demographic factors would help in analyzing and generalising the susceptibility results to other populations.

3. The reasoning for choosing to test nitroxoline susceptibility with agar dilution over broth microdilution is not really explained. BMD is the method recommended for other antibiotics for *Aerococcus*, and has been used in other research studies for testing susceptibility to nitroxoline for other microorganisms.

4. There is no statement regarding an approval for the study by an ethical review board. The authors state that active consent from the patients whose samples were studied is not needed in their legal system. Is there no need for an ethical review of the study protocol when accessing the clinical samples? If not, please state so in the manuscript.

Minor comment:

5. The introduction does not really touch upon the role of *Aerococcus* as an endocarditis pathogen. Previous studies indicate that the risk of endocarditis is rather high when finding *Aerococcus* in blood culture (similar to the risk of endocarditis when finding non-beta-hemolytic streptococci in blood). While this is not the focus of this article, it is an important manifestation of invasive *Aerococcus* infections, and should probably be mentioned.

Reviewer #4 (Comments for the Author):

This manuscript describes the action of nitroxoline on the two aerococcal species *Aerococcus urinae* and *Aerococcus sanguinicola*. Nitroxoline has the potential to be widely used against lower UTI and aerococci have emerged as potentially important uropathogens. Therefore, the study is of interest. The manuscript is relatively well-written. I have comments and suggestions for improvement below.

Comments

1. If the activity of nitroxoline is "excellent" or "poor" is not really shown. What is shown is that MICs are typically under the suggested *E. coli* breakpoint. Exchange "excellent" and "poor" at least in abstract and results.
2. Abstract it is claimed that "alternative oral drug to eradicate *A. urinae* from the urinary tract". I think that the reason to use the drug is to treat lower UTI, not to eradicate. Please change.
3. Importance "while ciprofloxacin was more often resistant". It is the bacterium and not the drug that is resistant. Please rephrase.
4. Introduction, please exchange references 4 and 5 to studies describing larger case series. It is not acceptable to refer to case reports when there are relatively large cohort studies on the topic. Please exchange.
5. Line 66, ciprofloxacin is an option for febrile UTI, change to "lower UTI".
6. Line 77, streptococci and enterococci should be in lower case and without italics. Only genus or species names should be in italics.
7. Line 89, please say what cut-off was used for species determination.
8. Line 92, "Susceptibility of ampicillin" should be "susceptibility to ampicillin".
9. Line 114, mention the figures for the two species separately.
10. Line 117, "uUTI" please introduce abbreviation.
11. Line 118 "almost every isolate with high nitroxoline MICs". Please say instead what the figures were and give significance of difference.
12. Line 121-123 this sentence is redundant. Mention instead how isolates would be classified according to zone diameter (EUCAST breakpoint 15 mm for 30 µg disc) and if there was any discrepancies between the two methods (MIC and disc-diffusion).
13. At line 134 it is worth mentioning that the study by Oskoi and coworkers investigated UTI caused by aerococci.
14. Line 151 "excellent correlation" is an overstatement. As suggested above an analysis of correlation in determination of SIR between the two methods is needed.
15. Line 153, I am not sure that *A. sanguinicola* and *A. urinae* are very similar at the genetical level. A reference would be needed to support this.
16. Line 170-171, this is an overstatement. This study at best suggest virulence mechanisms.
17. Table 1, zone sizes should be presented. Possibly as a supplemental table or figure.
18. Line 320 "adjusted from" is not correct, exchange to "for".

Staff Comments:

Preparing Revision Guidelines

Please return the manuscript within 60 days; if you cannot complete the modification within this time period, please contact me. If you do not wish to modify the manuscript and prefer to submit it to another journal, please notify me of your decision immediately so that the manuscript may be formally withdrawn from consideration by Microbiology Spectrum.

Ahmadzada et al. describe susceptibility differences between *Aerococcus urinae* and *Aerococcus sanguinicola* through disk diffusion and agar dilution. They identify a difference in susceptibility to nitroxoline and to a lesser extent ciprofloxacin between the species and propose a clinical use of nitroxoline in uncomplicated urinary tract infections.

Minor Comments:

1. Since a difference in susceptibilities was noted between *A. urinae* and *A. sanguinicola*, did you investigate the different patient populations? Were the individuals with *A. sanguinicola* treated with more antibiotics than the patients with *A. urinae*? This would help understand if the observed differences are intrinsic resistance vs. recent acquisitions (additional support for statement in Line 140)
2. Line 114 – for ciprofloxacin resistance, further discussion about observation in both *A. urinae* and *A. sanguinicola* could be beneficial.
3. As the authors note (Line 45), there is a need for additional susceptibility data in the literature. Could the authors add additional information on how the patients with *Aerococcus* sp. were treated? Is there correlation with the drugs that they studied and what was used for treatment?
4. Line 145 – to improve clarity, it would be helpful to compare and contrast methodology between the current submission and Scholtz et al.
5. Line 149 – “fluorochinolone” needs updated to “fluoroquinolone”
6. Discussion section should elaborate more on previous studies and how they are the same or different than present study (see comment 4).
7. See lines 150/151 – this paragraph is a single sentence and could be combined with 152 – 156.
8. Line 164 – comma not needed after “The fact”

Comments on “Susceptibility of *Aerococcus urinae* and *Aerococcus sanguinicola* to standard antibiotics and to nitroxoline”

The authors have performed AST on aerococcal isolates with a panel of antibiotics, including nitroxoline. To my knowledge, AST for nitroxoline and aerococci has not been reported before. There are some remarks and questions listed below.

L20 The authors write “..are emerging species..”. Maybe it is more correct to say something like, “...have been increasingly recognized as cause of UTI during the recent decade”.

L34 “all *A. urinae* isolates would be interpreted as susceptible”. This seems to be incorrect according to the result (where 97.6 % were S).

L47 “while ciprofloxacin was more often resistant”. This is incorrect. It is of course the isolates that are resistant, not the antibiotic.

L55-56 Same comment as on Line 20. The references used indicate that *Aerococcus* species have been recognized the last couple of years, but in fact have been increasingly recognized as cause of UTI during the recent decades. There are many references 5-15 years back that could be used also.

L59 I would suggest that there is a reference to the MALDI statement in the manuscript.

L64-66 A good and valid point.

L70 A better reference should be used to this statement.

L79 Administration instead of application?

L85 This section raises some questions. First, are the isolates in the study representative for all *A. urinae* and *A. sanguinicola* isolates? How many urine samples does the laboratory analyse each month/year? And what is the proportion of *Aerococcus* spp in the positive urine cultures? How were the samples selected – was it a random or a consecutive selection during the period? Which is the population base (size), is it both inpatient and outpatients? What were the gender and age distribution for all patients in the study? Was there any clinical information about symptoms?

L89 Which MALDI score was used as cut-off for reliable identification to the species level, and were all isolates identified according to this level?

L97 Reference 19 refers to Etest (EUCAST). Is that relevant for the method used?

L103 Temperature could also be specified.

L105 For what was the Mann-Whitney U test used?

L112 The sentence could be improved in line with “All aerococcal isolates tested **in this study** were susceptible to (Table 1).”

L119 The correlation coefficient (r-value) is not of great clinical interest. Also, there is a lack of discussion around what the result means and how it could be explained that the methods correlate rather poorly.

It would be of more value to see a histogram with MIC and zone diameter (“EUCAST model”) and a proposed cut-off zone diameter for each species that could be used to categorize the isolates as S or R (when using MIC 16 mg/L).

Dear reviewers,

Thank you very much for your time to review our manuscript and provide us with helpful comments, which have greatly improved the manuscript. We have revised our manuscript accordingly and hope that it is now acceptable for publication in Spectrum.

Response to reviewers' comments

Reviewer #1 (Comments for the Author):

1. Since a difference in susceptibilities was noted between *A. urinae* and *A. sanguinicola*, did you investigate the different patient populations? Were the individuals with *A. sanguinicola* treated with more antibiotics than the patients with *A. urinae*? This would help understand if the observed differences are intrinsic resistance vs. recent acquisitions (additional support for statement in Line 140)

We agree that this analysis would have been interesting, unfortunately we were not able to directly compare the different populations since we had no access to these data. We were not involved in the treatment planning of the patients. Patients' symptoms, past medical history and prescribed treatment were not available to us. The focus of the present article is on in vitro susceptibility on current antibiotics and on nitroxoline.

2. Line 114 - for ciprofloxacin resistance, further discussion about observation in both *A. urinae* and *A. sanguinicola* could be beneficial.

Resistance to ciprofloxacin was 16/166 (9,6%) in *A. urinae* and 4/18 (22,2%), in *A. sanguinicola*, $p=0.11$. This information was added in lines 121-122 of the revised version of the manuscript. Additionally, the results were compared more in detail to those of other studies.

3. As the authors note (Line 45), there is a need for additional susceptibility data in the literature. Could the authors add additional information on how the patients with *Aerococcus* sp. were treated? Is there correlation with the drugs that they studied and what was used for treatment?

Unfortunately, this information is not available to us.

4. Line 145 - to improve clarity, it would be helpful to compare and contrast methodology between the current submission and Scholtz et al.

The paragraph was modified for more clarification and the methodology is now discussed in more detail, as suggested (Line 154-163).

5. Line 149 - "fluorochinolone" needs updated to "fluoroquinolone"
done

6. Discussion section should elaborate more on previous studies and how they are the same or different than present study (see comment 4).

We agree with reviewer 1 and modified parts of the discussion (see comment 4).

7. See lines 150/151 - this paragraph is a single sentence and could be combined with 152 - 156.

#done

8. Line 164 - comma not needed after "The fact"

#done (see line 186)

Reviewer #2 (Comments for the Author):

The authors have performed AST on aerococcal isolates with a panel of antibiotics, including nitroxoline. To my knowledge, AST for nitroxoline and aerococci has not been reported before. There are some remarks and questions listed below.

L20 The authors write "..are emerging species..". Maybe it is more correct to say something like, "...have been increasingly recognized as cause of UTI during the recent decade".

We added this and thank the reviewer for this helpful comment (line 20, 21)

L34 "all A. urinae isolates would be interpreted as susceptible". This seems to be incorrect according to the result (where 97.6 % were S).

corrected (line 32-36)

L47 "while ciprofloxacin was more often resistant". This is incorrect. It is of course the isolates that are resistant, not the antibiotic.

corrected

L55-56 Same comment as on Line 20. The references used indicate that Aerococcus species have been recognized the last couple of years, but in fact have been increasingly recognized as cause of UTI during the recent decades. There are many references 5-15 years back that could be used also.

We agree with reviewer 2 and deleted the word "recently" from our introducing sentence for more clarification.

L59 I would suggest that there is a reference to the MALDI statement in the manuscript.

done

L64-66 A good and valid point.

We thank the reviewer for his comment.

L70 A better reference should be used to this statement.

New reference was added (Carkaci D et al. (2027) *Aerococcus urinae and Aerococcus sanguinicola*: Susceptibility Testing of 120 Isolates to Six Antimicrobial Agents Using Disk Diffusion (EUCAST), Etest, and Broth Microdilution Techniques)

L79 Administration instead of application?

was changed.

L85 This section raises some questions. First, are the isolates in the study representative for all *A. urinae* and *A. sanguinicola* isolates? How many urine samples does the laboratory analyse each month/year? And what is the proportion of *Aerococcus* spp in the positive urine cultures? How were the samples selected - was it a random or a consecutive selection during the period? Which is the population base (size), is it both inpatient and outpatients? What were the gender and age distribution for all patients in the study? Was there any clinical information about symptoms?

In the study period (12/2016-6/2018) 30718 urine samples from 17570 patients were analyzed. Of these, 312 grew *A. urinae/sanguinicola*. After removing duplicate isolates, 278 isolates remained. As susceptibility testing of *Aerococcus* has only recently been introduced, all isolates that were identified and assessed for susceptibility testing in clinical routine diagnostics were stored at -80°C in glycerol stocks as part of a quality control measure. The isolates were selected as part of a consecutive selection during the study period. However not all remaining isolates could be included in the study due to the retrospective nature (some isolates did not grow anymore, or were not stored initially). Overall, 184/278 (66.2%) isolates were available for further analysis. Therefore, our isolate collection can be considered representative for our patient population. Nevertheless, we added the information that not all isolates of the study period were available for analysis as a limitation of the study (lines 177-179). Also, the lack of clinical data (as discussed in the limitations) may lead to overestimation of the uropathogenic potential of aerococci. However, the aim of our study was to investigate the susceptibility of standard of care antimicrobials and to assess the activity of nitroxoline *in vitro*.

We therefore rewrote this section completely and included further information on the patients as requested (lines 87-96).

L89 Which MALDI score was used as cut-off for reliable identification to the species level, and were all isolates identified according to this level?

#A score of 2.0 was used, this information was added (line 97-98)

L97 Reference 19 refers to Etest (EUCAST). Is that relevant for the method used?

The study includes also data on Broth Micro Dilution compared to Etest which is relevant for our study.

L103 Temperature could also be specified.

Temperature was added. (Line 112)

L105 For what was the Mann-Whitney U test used?

#We thank the reviewer for this comment. The Mann-Whitney U test was indeed not used but Fisher's exact test for comparison of categorical data. This information was added accordingly.

L112 The sentence could be improved in line with "All aerococcal isolates tested in this study were susceptible to (Table 1)."

#was changed. (Line 120)

L119 The correlation coefficient (r-value) is not of great clinical interest. Also, there is a lack of discussion around what the result means and how it could be explained that the methods correlate rather poorly.

It would be of more value to see a histogram with MIC and zone diameter ("EUCAST model") and a proposed cut-off zone diameter for each species that could be used to categorize the isolates as S or R (when using MIC 16 mg/L).

We agree with the reviewer and added a histogram to the manuscript (Figure 1). The scattergrams were moved to the supplement.

Reviewer #3 (Comments for the Author):

The work that the authors have done in testing the susceptibility of *Aerococcus* to nitroxoline is interesting. There are however some potential issues that I believe should be addressed.

1. The collection of the studied samples is not clearly described. Were all findings of *Aerococcus urinae* and *A. sanguinicola* from 2016 to 2018 included in the study? If not, was there any possibility of a systematic bias in which samples were included or not? What percentage of samples identified during this period was included? Were all samples identified at the same laboratory?

This topic was also addressed by reviewer two and we changed the paragraph for the isolates in the method section (lines 87-96) accordingly. Overall, 184/278 (66.2%) of *Aerococcus* isolates were available for the study, some isolates were either not stored initially or did not grow after thawing. However, we could not identify any systematic bias. All isolates were identified in the same laboratory, using the same identification method (Bruker MALDI-ToF) and some susceptibility testing methods.

2. The patient population that the urinary samples were taken from is not described. Do the laboratory/ies in the study take samples from primary health care centers as well as secondary and tertiary care centers? From public or private clinics? This will potentially affect the population (with its accompanying microbiome) that is included in a study such as this. How large is the catchment area of the microbiology laboratory with its associated hospitals and other clinics? Knowledge of these demographic factors would help in analysing and generalising the susceptibility results to other populations.

Isolates were from patients of the University Hospital of Cologne, a large tertiary care center located in the western part of Germany. Most samples were from inpatients (121/184; 65.7 % of isolates); 93/184 (50.5%) were female and the median age was 77 years. The Institute for Medical Microbiology does not receive samples from primary health care centers or private clinics. We agree with the reviewer that this could have an impact on the population. For that reason, we expanded the previous limitation, now reading "Since most isolates were from inpatients from a tertiary care centre, additional risk factors may be overrepresented compared to uncomplicated UTI for which nitrofurantoin is approved. Additionally, our patient population might not be comparable to other population, e.g. from primary care which could have an impact on the overall susceptibility."

3. The reasoning for choosing to test nitrofurantoin susceptibility with agar dilution over broth microdilution is not really explained. BMD is the method recommended for other antibiotics for *Aerococcus*, and has been used in other research studies for testing susceptibility to nitrofurantoin for other microorganisms.

We thank the reviewer for this comment. For broth microdilution of *Aerococcus* spp., Mueller Hinton broth with 5% horse blood is usually used, which works fine for beta-lactam antibiotics. Unfortunately for nitrofurantoin this does not produce easy to read and reproducible results. Similarly, EUCAST recommends agar

dilution for fluoroquinolone susceptibility testing in aerococci as this method may produce clearer endpoints (EUCAST breakpoint table, version 12.0).

We added a brief comment about this to the discussion section (168-173)

4. There is no statement regarding an approval for the study by an ethical review board. The authors state that active consent from the patients whose samples were studied is not needed in their legal system. Is there no need for an ethical review of the study protocol when accessing the clinical samples? If not, please state so in the manuscript.

Isolates were grown as part of routine diagnostics. As this work is based only on isolates (containing no patient material) and no additional analyses on patient samples were carried out in this study, no patient consent or further ethical clearance is required according to the regulation of the University of Cologne. This information was added in lines 325-329.

Minor comment:

5. The introduction does not really touch upon the role of *Aerococcus* as an endocarditis pathogen. Previous studies indicate that the risk of endocarditis is rather high when finding *Aerococcus* in blood culture (similar to the risk of endocarditis when finding non-betahemolytic streptococci in blood). While this is not the focus of this article, it is an important manifestation of invasive *Aerococcus* infections, and should probably be mentioned.

#The role of *Aerococcus* in endocarditis was added in the introduction.

Reviewer #4 (Comments for the Author):

This manuscript describes the action of nitroxoline on the two aerococcal species *Aerococcus urinae* and *Aerococcus sanguinicola*. Nitroxoline has the potential to be widely used against lower UTI and aerococci have emerged as potentially important uropathogens. Therefore, the study is of interest. The manuscript is relatively well-written. I have comments and suggestions for improvement below.

Comments

1. If the activity of nitroxoline is "excellent" or "poor" is not really shown. What is shown is that MICs are typically under the suggested *E. coli* breakpoint. Exchange "excellent" and "poor" at least in abstract and results.

#done

2. Abstract it is claimed that "alternative oral drug to eradicate *A. urinae* from the urinary tract". I think that the reason to use the drug is to treat lower UTI, not to eradicate. Please change.

#done (line 39)

3. Importance "while ciprofloxacin was more often resistant". It is the bacterium and not the drug that is resistant. Please rephrase.

#done (line 48)

4. Introduction, please exchange references 4 and 5 to studies describing larger case series. It is not acceptable to refer to case reports when there are relatively large cohort studies on the topic. Please exchange.

The references were changed. (lines 222-229. Senneby et al (2012), Sunnerhagen et al (2016), Astudillo et al (2003))

5. Line 66, ciprofloxacin is an option for febrile UTI, change to "lower UTI".

#done (line 67)

6. Line 77, streptococci and enterococci should be in lower case and without italics. Only genus or species names should be in italics.

#done (line 78)

7. Line 89, please say what cut-off was used for species determination.

#done (line 98)

8. Line 92, "Susceptibility of ampicillin" should be "susceptibility to ampicillin".

#done (line 101)

9. Line 114, mention the figures for the two species separately.

#The figures were moved to the supplement and the figures were mentioned separately.

10. Line 117, "uUTI" please introduce abbreviation.

#done (line 74)

11. Line 118 "almost every isolate with high nitroxoline MICs". Please say instead what the figures were and give significance of difference.

This line was rephrased as follows: If the EUCAST nitroxoline breakpoint for *E. coli* and uncomplicated UTI was applied (16 mg/L), 97.6% (162/166) of *A. urinae* isolates would be interpreted as susceptible, compared to 0% (0/18) of *A. sanguinicola* isolates, $p < 0.0001$. (line 125)

12. Line 121-123 this sentence is redundant. Mention instead how isolates would be classified according to zone diameter (EUCAST breakpoint 15 mm for 30 µg disc) and if there was any discrepancies between the two methods (MIC and disc-diffusion).

We added a histogram for more clarification and discussed the MIC and disk diffusion correlation/results in more detail in the discussion section (see also comments of reviewer 2). The following sentence was added: Similarly, if disk diffusion inhibition zones were categorized using the EUCAST nitroxoline breakpoint for *E. coli* (15 mm) 163/166 (98.1 %) of *A. urinae* isolates would be interpreted as susceptible and 3/166 (1,8%) as resistant.

13. At line 134 it is worth mentioning that the study by Oskoi and coworkers investigated UTI caused by aerococci.

#done (lines 142-145)

14. Line 151 "excellent correlation" is an overstatement. As suggested above an analysis of correlation in determination of SIR between the two methods is needed.

We agree on that, deleted "excellent" and included additional information as described within previous responses to the reviewers

15. Line 153, I am not sure that *A. sanguinicola* and *A. urinae* are very similar at the genetical level. A reference would be needed to support this.

We thank the reviewer for this comment and deleted the statement

16. Line 170-171, this is an overstatement. This study at best suggest virulence mechanisms.

This part was deleted.

17. Table 1, zone sizes should be presented. Possibly as a supplemental table or figure.

We thank the reviewer for this comment, however we don't think this is necessary. For almost all isolates/ drugs the susceptibility rate is 100% and the results are in line with other *in vitro* studies about commonly used drugs for treatment of aerococcal urinary tract infection. Our study focuses on the novel data on nitroxoline and the other drugs are displayed as comparators only. Also, some of the presented data of figure one is based on clinical routine testing and zone diameters were not documented for all isolates. Due to the high rates of susceptibility in line with other studies we did not repeat susceptibility testing of all routine antibiotics within our study.

18. Line 320 "adjusted from" is not correct, exchange to "for".

was changed (line 365)

February 5, 2023

Prof. Axel Georg Hamprecht
University of Oldenburg
Institute for Medical Microbiology and Virology
Brandenburger Str. 19
Oldenburg 26133
Germany

Re: Spectrum02763-22R1 (Susceptibility of *Aerococcus urinae* and *Aerococcus sanguinicola* to standard antibiotics and to nitroxoline)

Dear Prof. Axel Georg Hamprecht:

Your manuscript has been accepted, and I am forwarding it to the ASM Journals Department for publication. You will be notified when your proofs are ready to be viewed.

Sincerely,

Rebekah Martin
Editor, Microbiology Spectrum
